# Scattering of Ultrashort X-ray Pulses by Various Nanosystems

**DOI:** 10.3390/nano10071355

**Published:** 2020-07-10

**Authors:** Marat Eseev, Andrey Goshev, Dmitry Makarov

**Affiliations:** Northern (Arctic) Federal University named after M.V. Lomonosov, Severnaya Dvina Emb. 17, 163002 Arkhangelsk, Russia; m.eseev@narfu.ru (M.E.); agoshev@hotmail.com (A.G.)

**Keywords:** ultrashort pulses, scattering, nanosystems, attosecond pulses, X-ray diffraction analysis, carbon nanotubes

## Abstract

Currently, the study of the scattering of ultrashort X-ray pulses (USPs) by various objects is an urgent task, in connection with the creation of powerful sources of USP generation. In this paper, the theory of the scattering of attosecond pulses by polyatomic structures is developed taking into account the magnetic component of USPs. It is shown that the scattering spectra depend not only on the structure of the target, but also on other characteristics of USPs. Results are presented of the calculation of the scattering spectra on various nanosystems, such as rings, groups of rings, carbon nanotubes (CNTs), and groups of co-directed CNTs (forest CNTs). The calculation results are presented in an analytical form, which allows a general analysis of the expressions. It was found that taking the magnetic component of the momentum into the scattering spectra into account leads to the generation of the second harmonic. In this case, the spectra have characteristic features and differ from the scattering spectra at the carrier frequency, which can complement ultra-high-resolution X-ray analysis. It is shown that the scattering spectra of some structures, for example, forest CNTs, have a strictly specified radiation direction and such material in the field of such USPs is non-reflective (completely black).

## 1. Introduction

In the last two decades, the generation of isolated attosecond pulses through the generation of high harmonics has provided a powerful tool for studying many important physical processes on the attosecond timescale [1]. Indeed, there is a tendency to increase the power of ultrashort pulses (USPs) of the electromagnetic field and reduce their duration [2,3,4,5]; for example, today the shortest pulse duration is 43 attoseconds (as) [6]. Research is being actively conducted and the technique of free electron X-ray lasers (XFEL) is being improved [5]. At present, a subfemtosecond barrier/border with a high peak power has been achieved, which makes it possible to study excitation in the molecular system and the motion of valence electrons with high temporal and spatial resolution [7]. The technique of generating ultrashort synchrotron pulsed radiation sources is also being actively developed [8]. This area of research is attracting considerable interest within the scientific community. It should enable a deeper understanding of the fundamental processes occurring in atoms and molecules [9], and greatly enhance the study of biomolecules, nanosystems, and complex polyatomic structures [10], as well as of various dynamic processes [11,12]. Furthermore, these new techniques enable the stages of a chemical reaction, and the generation of higher harmonics [3,4] to be traced. Moreover, at present there is a technical possibility to produce various complex polyatomic systems. For example, it is possible to obtain various nanotubes [13] and nanocomposites [14]. Two-dimensional materials can be obtained: graphene [15], borophene [16] and others.

When considering effects associated with the interaction of an electromagnetic field with atoms, the influence of the magnetic component of the pulse is usually neglected due to its smallness. Indeed, it is well known that the force acting on a charged particle from the side of the magnetic component of the electromagnetic field is approximately c = 137 atomic units (a.u.) times less than its electric component. However, when considering some phenomena, the contribution of this force is very significant and neglecting it can lead to a misunderstanding of certain processes: for example, in [17], when considering tunneling ionization in the high-field regime, its contribution was found to be very significant; and in [18], taking into account the magnetic component of USPs made it possible to detect the orientation effects of molecular anions. 

Various approaches are used to describe the scattering of USPs of an electromagnetic field. From a quantum mechanical point of view, light-scattering processes (both elastic and inelastic scattering) are usually explained using the first or next order of perturbation theory [19,20,21,22,23,24,25]. The semiclassical approach is also used to describe the scattering of light [26,27,28] this is when the atomic system is considered to be quantum, and the electromagnetic field is classical. More recently, studies have begun to appear that show that when using USPs in an X-ray diffraction analysis of a substance, the usual approaches may not be correct (see [12] and the references therein). For example, in [29] the quantum theory of USPs was developed in a general form, which shows that in the case of X-ray diffraction analysis of polyatomic systems, it is necessary to take into account the quantum nature of the scattered USPs.

In this paper, the theory of the scattering of attosecond pulses on polyatomic structures has been developed taking into account the magnetic component of USPs, and studies [29] and [30] have been used as a basis. In [29], a theoretical apparatus had been developed in a general form for calculating the scattering spectra of USPs. This theory used an unknown wave function of electrons in an USP field, which was found in [30]. Thus, a natural continuation of the development of these approaches is both their generalization and specific results obtained on the scattering spectra of polyatomic systems. The work presented here is a generalization in which analytical expressions for the scattering spectra of specific multi-atoms systems are obtained. The main parameters are found that are responsible for the characteristic diffraction pattern with which it is possible to carry out X-ray diffraction analysis of complex polyatomic structures.

The atomic system of units is used throughout: $\hbar =1,|e|=1,m_{e}=1$, where ℏ is “reduced” Planck’s constant, e is the electron charge, and $m_{e}$ is electron mass. 

## 2. Scattering Ultrashort Pulse (USP)

Consider a multi-electron atom which interacts with plane waves of USP extending towards the direction $n_{0}$. The duration of such a pulse is τ will be considered much shorter than the characteristic atomic time $\tau_{a}∼1$, i.e., ττa≪1, which allows us to consider the radiation process in the framework of the theory of sudden perturbations. In this approximation, the intrinsic Hamiltonian of the system can be neglected, since the electron in the atom does not have time to evolve under the action of the field of the atom due to the interaction of the momentum with the electron in the atom being too fast. It was shown in [30] that, in this approximation, taking into account the magnetic component of the electromagnetic field, the wave function of the atomic electron will be
(1)$$\psi (t)=\varphi_{0}\left(x,y,z+\int_{-\infty }^{t}\frac{Er}{m*c}\frac{1}{1+\frac{Er}{m*c^{2}}}dt\right)\cdot \frac{1}{\sqrt{\left|1+\frac{Er}{m*c^{2}}\right|}}\cdot \exp\left[-i\int_{-\infty }^{t}Er\frac{1+\frac{Er}{2m*c^{2}}}{1+\frac{Er}{m*c^{2}}}dt\right],$$
where $m^{*}$ is the effective electron mass; r is the radius vector of the electron in the selected coordinate system; E is the magnitude of the electromagnetic field strength in USP [30]; c is the speed of light; $\varphi_{0}(r)$ is the initial wave function of an electron in an atom. It should be added that Equation (1) has the necessary property of completeness in quantum mechanics, regardless of E in USP. There are also limitations under which Equation (1) is applicable: the characteristic frequency USP $\omega_{0}$ should be in the region of X-ray frequencies, and the duration of the USP should be less than the characteristic atomic time ττa≪1. Although the condition ττa≪1, as discussed in [30], is not rigid enough to use Equation (1). We are interested in electromagnetic fields that are not so strong as to take into account relativistic effects. In this case, decomposing in the Maclaurin series over a small parameter Ec2≪1, we obtain
(2)$$\psi (t)=\varphi_{0}\left(r\right)\exp\left[-i\int_{-\infty }^{t}\left(Er-\frac{1}{2}{\left(\frac{Er}{c}\right)}^{2}\right)dt\right].$$

Let some atom be located at the origin of the coordinate system, and the entire polyatomic system be in the ground state $|\varphi_{0}\rangle =|\varphi_{0}(r_{1},r_{2}\ldots r_{N})\rangle$. Furthermore, we will use the results of [29] and the wave function given in Equation (2). As a result, the probability W of producing a photon of a given frequency ω per unit solid angle $\Omega_{k}$ with the simultaneous transition of a polyatomic system from the ground state to all possible final states will be:(3)$$\frac{d^{2}W}{d\omega d\Omega_{k}}=\frac{1}{{\left(2\pi \right)}^{2}}\frac{1}{c^{3}\omega }\langle \varphi_{0}|{\left|\sum_{a,e}\exp(-ikR_{a,e})f(r_{a,e})\right|}^{2}|\varphi_{0}\rangle ,R_{a,e}=R_{a}+r_{a,e},$$
where $R_{a,e}$ are the coordinates of the electron of the atom *a* relative to the selected coordinate system; $r_{a,e}$ are the coordinates of the electron relative to the atom number *a*, ω-photon frequency; and n=kk is the unit vector along the photon emission direction. Equation (3) describes the scattering spectrum summed over the polarizations of the photon. In Equation (3) $f(r_{a})=\left[\tilde{E}(\omega )\times n\right],$
(4)$$\tilde{E}(\omega )=\int_{-\infty }^{+\infty }\left(E(r_{a,e},t)-\frac{1}{2}\nabla_{a}{\left(\frac{E(r_{a,e},t)}{c}\right)}^{2}\right)e^{i\omega t}dt,$$
where ∇a,e=∂∂ra.e.

For example, in the particular case of a monoatomic system and a hydrogen-like atom, the same result was obtained in [29]. Summation in Equation (3) goes over all the atoms *a* of which the polyatomic system consists and over all the electrons *e* that enter these atoms. Equation (3) is of a general nature; however, it is difficult to calculate the scattering spectra using it directly, because this calculation requires knowledge of the wave function of a multi-electron atomic system in its initial state $|\varphi_{0}\rangle =|\varphi_{0}(r_{1},r_{2}\ldots r_{N})\rangle$. Carrying out such numerical calculations in the case of a polyatomic system is practically an unsolvable task even on modern supercomputers. Large calculations can be avoided by expressing the average in Equation (3) in terms of the spatial density of atomic electrons ρ(r). Such an approach is well known: in [31], for example, it was used to calculate the spectra of USP scattering by a single atom. We will consider that the studied polyatomic system comprises identical atoms. Having performed calculations similar to those in [31], we obtain:(5)$$\frac{d^{2}W}{d\omega d\Omega_{k}}=\frac{1}{{\left(2\pi \right)}^{2}}\frac{1}{c^{3}\omega }\left(N_{a}N_{e}S(\omega ,n,n_{0})+N_{e}^{2}\delta_{N}(p)F(\omega ,n,n_{0})\right),$$
where $S(\omega ,n,n_{0})=G(\omega ,n,n_{0})-F(\omega ,n,n_{0})$, $N_{a}$ is the total number of atoms in the system, $N_{e}$ is the number of electrons in an atom, and $G(\omega ,n,n_{0})$ and $F(\omega ,n,n_{0})$ are the average values expressed in terms of electron density in the form:(6)$$G(\omega ,n,n_{0})=\frac{1}{N_{e}}\int \rho (r){\left|f(r)\right|}^{2}dr,F(\omega ,n,n_{0})=\frac{1}{N_{e}{}^{2}}{\left|\int \rho (r)f(r)e^{-ikr}dr\right|}^{2},$$
and the factor $\delta_{N}(p)$ (analyzed in a separate section), which completely determines the geometric arrangement of atoms in the target, is calculated in the general form:(7)$$\delta_{N}(p)=\sum_{a,b}e^{ip(R_{a}-R_{b})}={\left|\sum_{a}e^{ipR_{a}}\right|}^{2},$$
where $p=\frac{\omega }{c}(n-n_{0})=k-k_{0}$ and it has the meaning of the recoil momentum in the scattering of USPs. Electronic density ρ(r) can be taken in various models. We will use the Dirac–Hartree–Fock–Slater model [32], in which $\rho (r)=\frac{N_{e}}{4\pi r}\sum_{i=1}^{3}A_{i}\alpha_{i}{}^{2}e^{-\alpha_{i}r}$, where $A_{i},\alpha_{i}-$ tabular coefficients that specify the electron density in the atom [32]. Using this model, one can find equations for $G(\omega ,n,n_{0})$ and $F(\omega ,n,n_{0})$ in an analytical form. As a result, we obtain (details of the integrals calculation can be found in [31]):G(ω,n,n0)=[E0×n]2|F1|2+6(ωc)2|F2|2(E0c)4∑i=13Aiαi4[n0×n]2+2(E0c)2∑i=13Aiαi2{[E0×n]2c2|F2|2−ωc(E0n)(n0n)Im(F1F2)},F(ω,n,n0)=1(4π)2|∑i=13Aiαi2Ji(ω,n,n0)|2,
(8)$$\begin{array}{l} J(\omega ,n,n_{0})=\frac{4\pi F_{1}}{p^{2}+\alpha_{i}{}^{2}}[E_{0}\times n]+\frac{8\pi iF_{2}}{{\left(p^{2}+\alpha_{i}{}^{2}\right)}^{2}}\frac{{Ε}_{0}p}{c^{2}}[E_{0}\times n]-\\ -\frac{4\pi iF_{2}}{{\left(p^{2}+\alpha_{i}{}^{2}\right)}^{3}}\frac{\omega }{c}\left\{{\left(\frac{E_{0}p}{c}\right)}^{2}-4{\left(\frac{{Ε}_{0}p}{c}\right)}^{2}+{\left(\frac{\alpha_{i}E_{0}}{c}\right)}^{2}\right\}[n_{0}\times n]. \end{array}$$

In Equation (8) $E(t,r)=E_{0}v(x,\gamma ,\omega_{0})$, where $x=t-n_{0}r/c$; $n_{0}$ is unit vector along the USP; γ is parameter specifying the spectral width; $\omega_{0}$ is carrier frequency, i.e., $v(x,\gamma ,\omega_{0})$ sets the form of the USP; $F_{1}(\omega )=\int_{-\infty }^{\infty }\nu (x,\gamma ,\omega_{0})e^{i\omega x}dx$; $F_{2}(\omega )=\int_{-\infty }^{\infty }\nu^{2}(x,\gamma ,\omega_{0})e^{i\omega x}dx$. It should be added that the functions $F(\omega ,n,n_{0})$ and $G(\omega ,n,n_{0})$ in Equation (8) are responsible for the USP scattering spectra from specific atoms. In other words, each atom has its own unique function, $F(\omega ,n,n_{0})$ and $G(\omega ,n,n_{0})$, for a given USP. The $F(\omega ,n,n_{0})$ function differs from $G(\omega ,n,n_{0})$ in that it can change the amplitude of diffraction maxima depending on which atoms we are considering in the system. In a polyatomic system, if $F(\omega ,n,n_{0})\gg G(\omega ,n,n_{0})$, then the studied system coherently scatters the USP, while $G(\omega ,n,n_{0})\gg F(\omega ,n,n_{0})$ or $F(\omega ,n,n_{0})∼G(\omega ,n,n_{0})$, then both the coherent and incoherent parts of the USP scattering spectrum can be present in the scattering spectrum.

The scattering spectrum described by Equation (5) is determined using the functions $F_{1}(\omega )$ and $F_{2}(\omega )$. For this, we will consider USPs of a Gaussian form:(9)$$E\left(r,t\right)=E_{0}\exp\left(-\gamma^{2}{\left(t-n_{0}r/c\right)}^{2}\right)\cos\left(\omega_{0}t-k_{0}r\right),$$
where k0=n0ω0c, τ=1/γ is the pulse duration, and γ is the attenuation parameter in a Gaussian pulse. We assume that the USP is multi-cycle, i.e., ω0γ≫1, then:(10)$$F_{1}=\frac{\sqrt{\pi }}{2\gamma }\left\{e^{-{\left(\frac{\omega -\omega_{0}}{2\gamma }\right)}^{2}}+e^{-{\left(\frac{\omega +\omega_{0}}{2\gamma }\right)}^{2}}\right\},F_{2}=\frac{\sqrt{\pi }}{4\sqrt{2}\gamma }\left\{e^{-{\left(\frac{\omega -2\omega_{0}}{2\sqrt{2}\gamma }\right)}^{2}}+e^{-{\left(\frac{\omega +2\omega_{0}}{2\sqrt{2}\gamma }\right)}^{2}}+2e^{-{\left(\frac{\omega }{2\sqrt{2}\gamma }\right)}^{2}}\right\}.$$

Also, we give the value: (11)$$f(r_{a,e})=\exp(ik_{0}r_{a,e})\left\{\left(F_{1}(\omega )-F_{2}(\omega )\frac{E_{0}r_{a,e}}{c^{2}}\right)\left[{Ε}_{0}\times n\right]-iF_{2}(\omega )\frac{\omega }{2c}{\left(\frac{E_{0}r_{a,e}}{c}\right)}^{2}\left[n_{0}\times n\right]\right\}.$$

Thus, the scattering spectrum can be considered quite definite. Note that the scattering in this case is localized near frequencies $\omega_{0}$ and $2\omega_{0}$ with dispersion γ, which is a consequence of the decomposing of the wave Equation (1) in the Maclaurin series on a small parameter Ec2≪1. Taking into account the following amendments—as a result of which, obviously, higher harmonics will appear—will be relevant when considering the relativistic case. Therefore, in our case they are not taken into account because of their smallness.

In limiting cases, Equation (5) demonstrates that it is possible to distinguish both the incoherent and coherent parts of the spectrum in the scattering spectrum. In other words, the scattering spectrum can be $\propto N_{e}N_{a}$ (for example, with $c\gg \omega_{0}/c\gg 1$ and a small number of atoms in the system, so that $\delta_{N}(p)N_{e}^{2}F(\omega ,n,n_{0})\ll N_{a}N_{e}S(\omega ,n,n_{0})$).

This case corresponds to the scattering of USPs by electrons in atoms independently of each other. In the case of the coherent part of spectrum $\propto N^{2}{}_{e}N^{2}{}_{a}$ (for example, for sufficiently small $\omega_{0}/c$, one can obtain $\delta_{N}(p)=N_{a}^{2}$, for $N_{e}\gg 1$). This case corresponds to the scattering of USPs by electrons in atoms together. It can be seen from Equation (5) that in the general case it is impossible to separate the coherent and incoherent parts of the spectrum. It is also seen from Equations (5) and (7) that in the case of a polyatomic system, the last term in Equation (5) is the main contributor to the scattering spectrum. In addition, it is this term that is responsible for the diffraction pattern, because the factor $\delta_{N}(p)$ determines the geometric arrangement of atoms in the target. Using the $\delta_{N}(p)$ factor, it is not always possible to determine in which cases the diffraction pattern has maxima and minima. This is due to the fact that, in Equation (5), the $\propto F(\omega ,n,n_{0})\delta_{N}(p)$ scattering spectrum and the $F(\omega ,n,n_{0})$ function can “cut off” the maxima in the factor $\delta_{N}(p)$. Although, at certain parameters of the USP, where the $F(\omega ,n,n_{0})$ has a weak dependence on the direction of the n, the diffraction pattern is determined by the factor $\delta_{N}(p)$.

## 3. Scattering by Multi-Atomic Systems

Thus, Equation (5) is the angular distribution of the USP scattering spectra of a polyatomic system in which the electron density is given by the Dirac–Hartree–Fock–Slater model. Consider a ring consisting of $N_{a}$ identical atoms located along the circumference of a ring of radius *R*, at an equal distance from each other. We introduce a rectangular coordinate system so that the origin of the coordinate system is in the center of the ring, and the *x*, *y* axes lie in the plane of the ring. Then the radius vector defining the position of the atom with number *a* is:Ra=R(Cos[2πaNa]i+Sin[2πaNa]j),
where *R* is the radius of the ring, and ***i*,**
***j*** and ***k*** are unit vectors in the *x*, *y*, and *z* axes, respectively. Next, it is necessary to find function (7) i.e., responsible for the geometric arrangement of atoms in the target. It is clear that,
Rap=pR⋅Sin(θ)Cos(2πaNa−φ),
where θ is the angle between ***p*** and the normal to the plane of the ring, φ is the angle between the *x* axis and the projection of the vector ***p*** on the *xOy* plane, and the atom number *a* takes the values *a* = 1,2…$N_{a}$. Next, we consider the sum in Equation (7) $\sum_{a}e^{ipR_{a}}$. This quantity for $N_{a}\gg 1$ can be replaced by an integral $\sum_{a}e^{ipR_{a}}=\frac{N_{a}}{2\pi }\int_{0}^{2\pi }\exp\left(ipR\cdot sin(\theta )\cdot cos(x)\right)dx=N_{a}J_{0}\left(pR\cdot \sin\theta \right)$, where $J_{0}(x)$ is the Bessel function. As a result, for the ring we obtain the equation:(12)$$\delta_{N}(p)=N_{a}{}^{2}J_{0}{\left(\frac{\omega }{c}R\left|(n-n_{0})\times k\right|\right)}^{2}.$$

Let us evaluate the result. As already stated, the spectral density of radiation has two pronounced harmonics, as demonstrated from the form of the function F_2_, which includes the doubled frequency associated with taking into account the magnetic component of the incident USP. The pulse scattered by a nanostructure (carbon ring) is inhomogeneous in direction, as can be seen from Figure 1; Figure 2. Figure 1 shows the spectral density of radiation on the ring of carbon atoms for various spatial locations of the detector, which is given by the angles Θ and ϕ. The angle between the direction of incidence of the pulse and the axis of the ring is fixed at 45°.

Figure 2 shows that the scattering spectrum on the ring has the following features:The scattering spectrum at the two harmonics has quantitative and qualitative differences. The amplitude of the spectral density of radiation at a doubled frequency is much smaller than at the first harmonic.Scattering in the region ω_0_ has a pronounced direction coinciding with the direction of incidence of the pulse **n**_0_. Radiation near 2ω_0_, in contrast, has greater symmetry and is maximum in the direction orthogonal to the incidence **n**_0_, which is associated with taking into account the magnetic component of the USP. Therefore, by positioning the detector at different angles to the system, it is possible to separately register both the first and second harmonics.Orientational effects (the position of the normal of the ring with respect to **n**_0_) are weakly expressed in the case of high frequencies.Both parts of the spectrum are sensitive to the number of ring atoms.

Thus, the main part of the radiation falls on the carrier frequency ω_0_, and the emission spectrum has a local character, which is amplified, passing into the delta function with increasing radius of the system. As can be seen from Equations (5) and (12), the angles that specify the direction of the pulse fall relative to the normal to the ring are included only in factor $\delta_{N}(p)$, whose contribution is very insignificant in the case of high frequencies ω0c≥5. However, the situation changes in the case of low frequencies; here, the terms with $F(\omega ,n,n_{0})$ and $G(\omega ,n,n_{0})$ are weakly dependent on the frequency ω, and the contribution of $\delta_{N}(p)$ to the resulting scattering becomes more noticeable. In other words, function $F(\omega ,n,n_{0})$ weakly affects the value in the spectrum $\propto F(\omega ,n,n_{0})\delta_{N}(p)$ and does not “cut out” the maxima defining the factor $\delta_{N}(p)$. Figure 3 shows the scattering spectra, depending on the orientation of the axis relative to the incident pulse for $\omega_{0}/c\approx 5$.

Despite the fact that, technically, a full scattering spectrum is recorded in an experiment d2WdΩkdω, the distinguishing features associated with the spatial arrangement of atoms in this case are only entering the factor $\delta_{N}(p)$. It is this factor that is responsible for the geometry of the target and the recording of diffraction maxima in the experiment; its contribution is precisely what makes the difference in the scattering spectra of various nanostructures. Knowledge of this factor $\delta_{N}(p)$ is of particular interest, and therefore we will further consider various special cases where this factor can be found analytically. 

Below, we present the results of calculating the factor $\delta_{N}(p)$ for such systems as a ring, a group of rings, a plane from a group of rings (PGR), carbon nanotubes (CNTs), and groups of co-directed CNTs (forest CNTs). A graphical representation of these structures is shown in Figure 4. In general, the factor $\delta_{N}(p)$ is determined by Equation (7). For one ring of Na atoms, we obtained Equation (12), which can be generalized to the case of a group of axially symmetric rings, in this case
(13)$$\delta_{N}(p)={\left(\sum_{n=1}^{M}N_{n}J_{0}\left(\frac{\omega }{c}R_{n}\left|(n-n_{0})\times k\right|\right)\right)}^{2},$$
where $N_{n}$ is the number of atoms on the *n*th ring, $R_{n}$ is the radius of the nth ring, and *M* is the number of rings. 

Next, we consider a structure consisting of a group of axially symmetric rings whose centers are located on the plane (PGR). To do this, we consider a planar system formed by identical multiple rings lying in the same plane so that the centers of the rings are located at the nodes of the rectangular lattice lying in the *xOy* plane. We denote the lattice period along the *x* axis as d_1_, and the period is equal along the *y* axis as d_2_.Then, the radius vector defining the position of any atom with number *a* on a ring centered at the node with numbers *a_1_,a_2_* will be equal to:(14)Ra=(a1−1)d1i+(a2−1)d2j+Rn(cos(2πaNn)i+sin(2πaNn)j),
where the integers *a_1_*, *a_2_*, are the numbers of nodes in the lattice, such that the node with numbers *a*_1_
*=* 1, *a*_2_
*=* 1 is located at the beginning of the coordinate system.

Factor $\delta_{N}(p)$ can now be found for multiple rings whose centers are in the nodes of the rectangular lattice:(15)$$\delta_{N}(p)={\left(\sum_{n=1}^{M}N_{n}J_{0}\left(R_{n}\left|p\times k\right|\right)\right)}^{2}\prod_{i=1}^{2}{\left(\frac{Sin\left[L_{i}pd_{i}/2\right]}{Sin\left[pd_{i}/2\right]}\right)}^{2},$$
where $N_{n}$ is the number of atoms in a ring of radius $R_{n}$; $L_{1},L_{2}$ is the number of nodes on the *x*, *y* axes, respectively; and $d_{1},d_{2}$ is the lattice period. 

It is also possible to obtain a structure similar to a CNT. We set the CNT so that the radius vector is expressed as Ra=(m−1)dk+Rn(cos(2πaN)i+sin(2πaN)j), then
(16)$$\delta_{N}(p)={\left(\sum_{n=1}^{M}N_{n}J_{0}\left(R_{n}\left|p\times k\right|\right)\right)}^{2}{\left(\frac{Sin\left[Ldpk/2\right]}{Sin\left[pkd/2\right]}\right)}^{2},$$
where $N_{n}$ is the number of atoms in a ring of radius $R_{n}$, L is the number of planes with rings, and *d* is the step between the planes. Thus, it is possible to define both a single-layer *m* = 1 and a multi-layer *m* > 1 CNT. For forest CNTs, the $\delta_{N}$ factor is obviously determined
(17)$$\delta_{N}(p)={\left(\sum_{n=1}^{M}N_{n}J_{0}\left(R_{n}\left|p\times k\right|\right)\right)}^{2}\prod_{i=1}^{3}{\left(\frac{Sin\left[L_{i}pd_{i}/2\right]}{Sin\left[pd_{i}/2\right]}\right)}^{2},$$
where $N_{n}$ is the number of atoms in the ring of radius $R_{n}$, $d_{1}=d_{1}i,d_{2}=d_{2}j,d_{3}=d_{3}k,$
$L_{1},L_{2}$ is the number of nodes on the *x* and *y* axes, respectively, and $L_{3}$ is the number of planes perpendicular to the *z* axis. 

The analytical form of the factor $\delta_{N}(p)$ allows the analysis of scattering spectra. It can be seen from Equation (12) that there are maxima whose width (the solid angle at which scattering occurs) is determined by the expression $\frac{\omega_{0}}{c}R$: the larger this expression, the narrower the maximum. The general appearance of the diffraction pattern depends on the number of atoms in the ring. With an increase in the radius of the ring, and while maintaining the number of atoms, it undergoes change. Diffraction effects from neighboring atoms are reduced and the radiation is almost spherically symmetrical. Similarly, one can determine the behavior of the factor $\delta_{N}(p)$ in the case of the second harmonic. Obviously, the probability of radiation will be less than at the first harmonic, and the width of the maxima will be narrower.

Let us present graphically the results of calculating the factor $\delta_{N}(p)$ for the polyatomic systems considered above. We show that each system has its own distinctive features, which allows us to judge the structure of complex polyatomic systems.

The factor $\delta_{N}(p)$ for *M* = 1 (one ring) for different angles of incidence of the USP relative to the axis of symmetry of the ring is presented in Figure 5. Note that for this part of the spectrum at any angle of incidence of the USP, the spectrum is divided into the reflected and transmitted spectrum. Furthermore, note that in this case the classical law of reflection, “the angle of incidence is equal to the angle of reflection”, is fulfilled for it.

### 3.1. Scattering on Several Axially Symmetric Rings

For case m>1, the same regularities as for the ring are fulfilled; we only note that, in addition to orientational sensitivity, the scattering spectra are supplemented by sensitivity to the distribution of atomic density. A minimum of radiation occurs in the case of a uniform distribution of atoms along the rings. 

### 3.2. Scattering on Carbon Nanotube (CNT) and Plane from a Group of Rings (PGR)

The analysis of the factor for CNTs and the forest CNT shows that scattering occurs in such a way that there is practically no reflection. Moreover, the scattering occurs in almost one direction, coinciding with the direction of the incident USP. The more atoms in such systems, the greater the unidirectional scattering with decreasing spectral width. This type of scattering is more clearly expressed for the forest CNT. It should be added that this kind of scattering is well known in the case of the optical frequency range, see for example [33], and is already used in many technical fields (super-black coatings, blocking out light, etc.) In our work, an effect of this kind for USPs is first theoretically presented in the literature. In the case of a large number of atoms in the system, factor $\delta_{N}(p)$ at the second harmonic will be similar to that at the first harmonic, the only difference being that the width of the maxima will be narrower. In the case where the number of atoms is not large, the scattering spectrum will be determined not only by the factor, but also by the incoherent part of the scattering, i.e., the first and second terms of Equation (5). As an example, in Figure 6 we present graphically the scattering for two cases.

## 4. Discussion and Conclusions

In the framework of the method of sudden perturbations (taking into account the fact that the duration of USPs is much shorter than the characteristic atomic time ττa≪1), a general analytical expression has been obtained for the scattering spectra on various nanosystems. Taking into account the magnetic component of the incident USP leads to the generation of the second harmonic in the scattering spectrum. Equation (5) obtained by us can be used in ultra-high resolution X-ray diffraction analysis. Indeed, this expression is more accurate in comparison with the well-known methods of X-ray diffraction analysis, see for example, [34,35]. Equation (6) includes the coherent and incoherent parts of the scattering spectrum, and also takes into account the magnetic component of the USP. These two important additions are usually ignored in X-ray analysis. From Equation (5) it is not difficult to obtain in the particular case the most frequently used expression in X-ray diffraction analysis [12,34,35]. For this, it is necessary to take into account only the electric component of USPs and assume that all the electrons in the atoms of the substance emit coherently. In this case, from Equation (5) we obtain:(18)$$\frac{d^{2}W}{d\omega d\Omega_{k}}=\frac{F_{1}^{2}(\omega ){\left[E_{0}\times n\right]}^{2}}{{\left(2\pi \right)}^{2}c^{3}\omega }{\left|\int n(r)e^{-ipr}dr\right|}^{2},$$
where n(r) is the distribution of electron density of the entire atomic system. We pass to the scattering cross section according to the well-known expression $\frac{d\sigma }{d\omega }=\frac{1}{I}\frac{dW}{d\omega }\omega$ [36], where intensity $I=\frac{c}{4\pi }\int_{-\infty }^{\infty }E^{2}(r,t)dt$ and considering that the impulse is long enough $\omega_{0}/\gamma \gg 1$, the well-known equation for the cross-section of X-ray diffraction [34] can be obtained:(19)$$\frac{d\sigma }{d\Omega_{k}}=\frac{\sin^{2}\theta }{c^{4}}{\left|\int n(r)e^{-iqr}dr\right|}^{2},\sin^{2}\theta ={\left[E_{0}\times n\right]}^{2}/E_{0}^{2},\;q=\frac{\omega_{0}}{c}(n-n_{0}).$$

The theory presented here improves the accuracy of, and complements, X-ray diffraction analysis, including its use for deciphering the spectra of complex polyatomic systems. There are two aspects to our work which will advance this field of study: first, our development of a more accurate theoretical apparatus will allow for a more accurate explanation of the diffraction pattern; second, the fact that the scattering of USPs occurs not only at the fundamental frequency $\omega_{0}$, but also at a double frequency $2\omega_{0}$. The scattering at each frequency has its own diffraction pattern, which makes it possible to compare them to make a more detailed and qualitative analysis of the studied polyatomic systems. It should be added that, using Equation (5), one can study both stationary and dynamic systems; the latter encompass complex molecules, including biomolecules, where bonds break or form, as well as charge migration in peptides and biological systems. For this, it is necessary to replace in Equation (5) ρ(r) by, ρ(r,t) where *t* is the moment in time at which the USP acts on the system under study. Such an approach for replacing ρ→ρ(r,t) of a USP in a field is well known and is discussed in detail in the literature, see for example [12].

## Figures and Tables

**Figure 1 nanomaterials-10-01355-f001:**
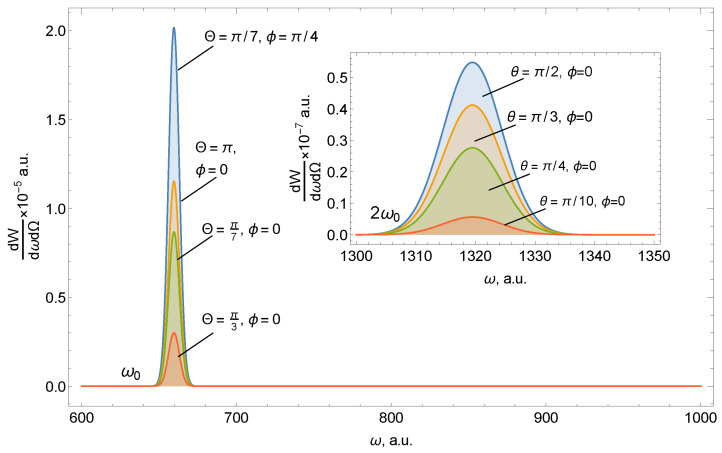
Spectral radiation density of ultrashort pulses (USPs) at the first ω_0_ = 660 and second 2ω_0_ harmonics (figure tab) depending on the position of the detector. The number of carbon atoms in the ring is Na = 6, the radius of the ring is R = 5, the pulse duration is τ = 43 as, and the field amplitude is E_0_ = 500. The angle of incidence of the pulse with respect to the axis of symmetry of the ring is α = π/4.

**Figure 2 nanomaterials-10-01355-f002:**
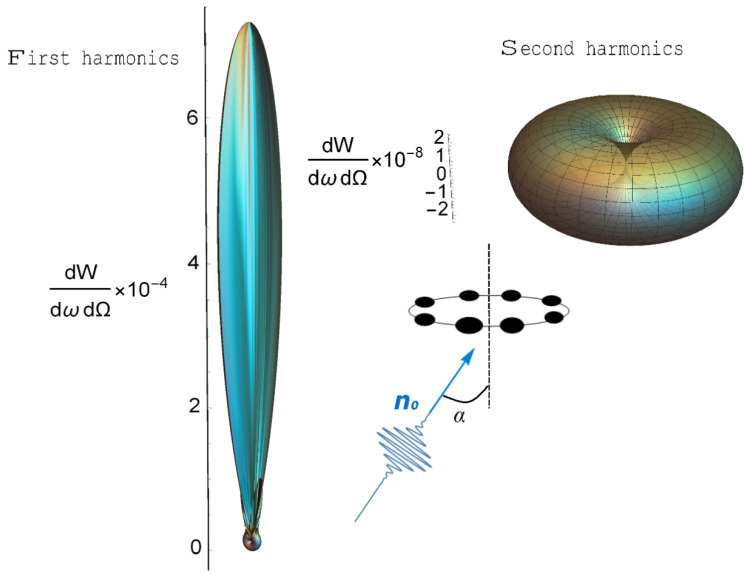
Three dimensional (3D) radiation pattern of USPs at the first ω_0_ = 660 and second 2ω_0_ harmonics. The number of carbon atoms in the ring is N_a_ = 6, the radius of the ring is R = 5, the pulse duration is τ = 43 as, and the field amplitude is E_0_ = 500, $\omega_{0}/c\approx 5$. The angle of incidence of the pulse with respect to the axis of symmetry of the ring is *α* = π/4. The *z* axis is in the direction of incidence of the pulse.

**Figure 3 nanomaterials-10-01355-f003:**
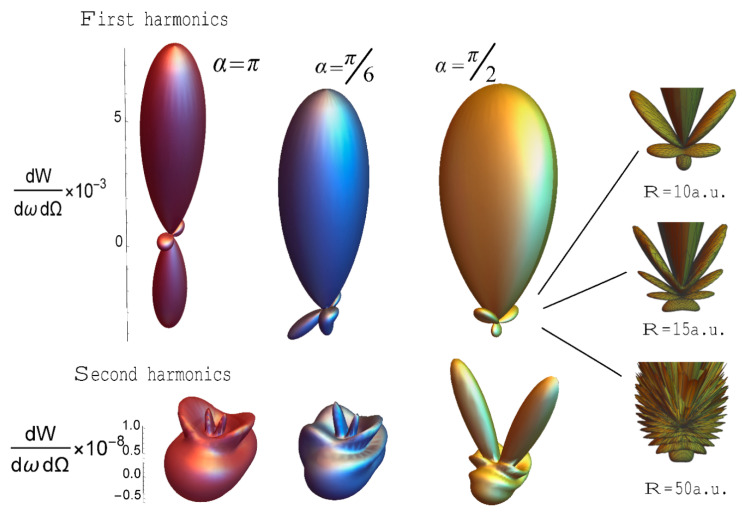
Spectral density of radiation depending on the orientation of the axis of the ring to the incident USP (angle *α*), at the first ω_0_ = 66 and second 2ω_0_ harmonics. For ω0c≈0,5, E_0_ = 500. The number of carbon atoms in the ring is R = 7, N_a_ = 6. The figure tab on the right shows the dependence of the factor $\delta_{N}(p)$ on the radius of the nanosystem (R).

**Figure 4 nanomaterials-10-01355-f004:**
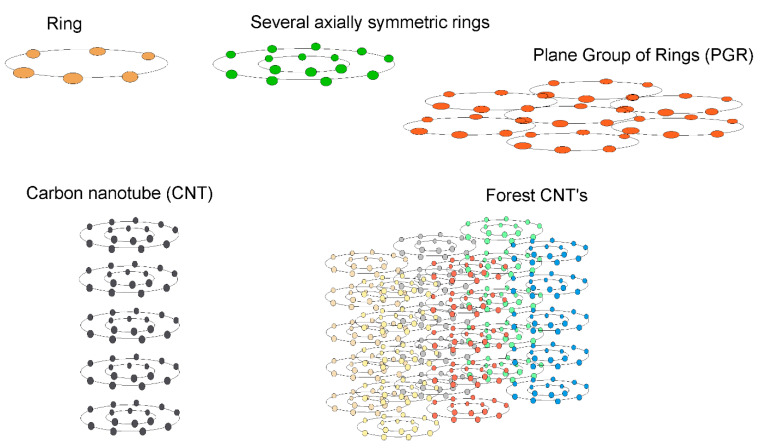
Image of the considered carbon nanostructures.

**Figure 5 nanomaterials-10-01355-f005:**
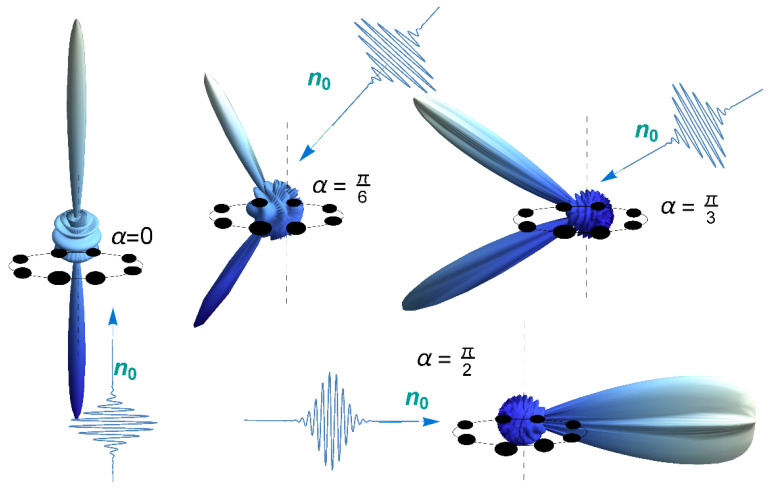
Orientational effects of the factor $\delta_{N}(p)$ depending on the angles of incidence of the momentum **n**_0_ to the normal of the ring. For a pulse duration of τ = 43 as, the field amplitude is E_0_ = 500, $\omega_{0}/c\approx 5$. Here **R** is the normal to the ring, **n**_0_ is the direction of the USP fall, α is the angle between the vectors ***n***_0_ and ***R***.

**Figure 6 nanomaterials-10-01355-f006:**
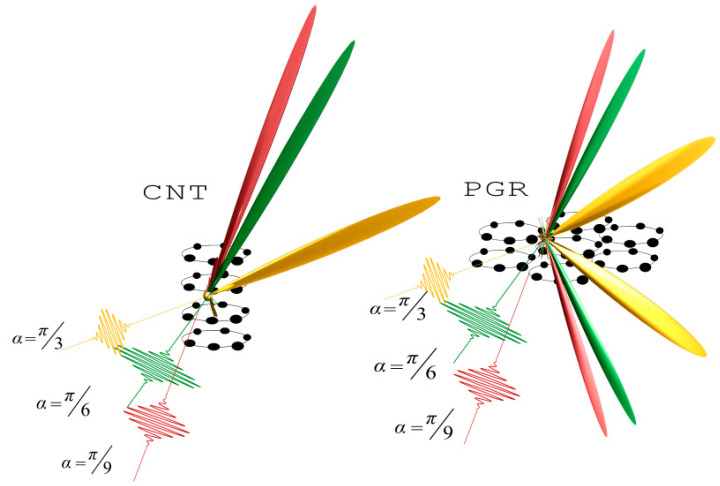
Orientation effects of factor $\delta_{N}(p)$ depending on the angles *α* of incidence of pulse ***n***_0_ to the carbon nanotubes (CNTs, parameters: R = 4, N = 6, L = 10) and plane from a group of rings (PGR, parameters: R = 3, N = 6, L_1_ = 6, L_2_ = 6). For a pulse duration of τ = 43 as, the field amplitude is E_0_ = 500, and $\omega_{0}=330$.

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
