# Peer review of "Scattering of Ultrashort X-ray Pulses by Various Nanosystems"

_nanomaterials, 2020, doi:10.3390/nano10071355_

Round 1
Reviewer 1 Report
# Re: Scattering of ultrashort X-ray pulses by various nanosystems
## by: M Eseev and A Goshev and D Makarov
In the manuscript, the authors apply the approach of sudden perturbations to theoretically derive spectra of ultrashort X-ray pulses scattered by nanosystems including multiple rings and carbon nanotubes. The approximation allows the authors to drop the own Hamiltonian of the scatterer and consider only the interaction energy. The main contribution of the manuscript seems to stem from taking into consideration also the magnetic field that, as authors claim, is often neglected in the literature. This provides extra accuracy to the computations and also allows computing non-negligible scattering spectra at the second harmonics.
The manuscript is consistently written, the models and methods are clearly outlined in a well-structured and educative manner with proper referencing of the literature. The mathematical derivations are not presented due to a rather short length of the manuscript, so it is difficult to evaluate their validity. Analytical results are derived, illustrated by sufficient figures, and discussed in a satisfactory manner.
The manuscript builds upon the previous works of the authors that are published in peer-reviewed journals. Particularly, Eq.(1), central to the subsequent derivations is borrowed from literature without lengthy explanations. Besides this, the present manuscript leaves an impression of a consistent complete study that contains enough new science to be published in a journal.
### Criticisms (Scientific)
- Sec. 2 starts from an approximation where the intrinsic Hamiltonian is dropped. It would be beneficial for the manuscript if the validity of this approximation was discussed in a few sentences.
- The quantities entering Eq. (1): E, r , m, the coordinate system, should be introduced.
- I personally would appreciate in the manuscript a brief excursion into the state of the art of the fabrication of the nanosystems considered. A few references concerning e.g. carbon nanotubes would suffice.
### Presentation
- In the figures 'harmonica' -> 'harmonics'
- In some figures, certain labels seem to be a bit too small. Generally, it would be appreciated if all labels throughout the manuscript had a consistent size.
- Possibly Sec.2 could have a more informative title, though this is opinionated.
- I would suggest expanding the acronym PGR within the panel of Fig.4 as 'Plane Group of Rings (PGR)'
### Minor
I find the wording in certain places suboptimal, in particular (but likely not limited to):
- In the abstract, 'X-ray pulses ... *of an electromagnetic field*' sounds like unnecessary repetition.
- p.1 l.25-29: the standard abbreviation for attosecond seems to be 'as', it can be introduced after the first occurrence of the word.
- p.1 l.30-32: the sentence seems to miss the verb/predicate.
- p.1 l.33: 'actively developing' -> 'being actively developed'
- p.3 l.82: 'on a small parameter' -> 'over a small parameter'
- p.3 l.86: perhaps 'probability of producing a photon W' -> 'probability W of producing a photon', otherwise I do not understand the meaning of this clause.
- p.3 'is the coordinates' -> 'are the coordinates'
- p.3 l.99 'directly' repeated x2.
- p.4 l.105 'consider the' -> 'consider that the'
- p.4 l.107 'similar calculations' -> 'calculations similar'
- p.4 l.123 'makes sense' -> 'has the meaning'
- p.6 l.172 'amount' -> 'quantity'
- p.7 l.190 'at two harmonics' -> 'at the two harmonics'
- p.7 l.214 'are only enter in a' -> 'are only entering the'
- p.11 l.288 'fall, is the angle' -> likely 'fall, alpha is the angle'
On an unrelated note, It seems likely that the authors use Mathematica to produce figures with labels that contain extensive mathematical notation. I would suggest (not necessarily for the present manuscript, but rather for the future works) using also the free package MaTeX that allows using LaTeX to typeset the labels.
Reviewer 2 Report
In this manuscript, the authors develop an analytical method to calculate the scattering spectra of attosecond pulses by polyatomic structures. This is especially important because, with the advent of powerful ultrashort electromagnetic field sources, the problem of studying the scattering of such pulses in matter is particularly relevant.
On the whole, this manuscript presents valuable new more accurate theoretical apparatus for a more accurate explanation of the diffraction pattern and new insights into the scattering of ultrashort X-ray pulses. The work presented here is a generalization of their published studies[Opt. Express 2019, 27(22), 31989; Opt. Lett. 2019, 44 (12), 3042; Journal of Experimental and Theoretical Physics, 2017, 125(2), 189]. The analysis of model and derivation of general expressions appears to be rigorous and sound. This is an interesting contribution, the analysis for CNTs and the forest CNT shows that scattering occurs in such a way that there is practically no reflection which would be quite unusual, and while there is not experimental evidence for this effect yet. So it may be publishable.
There are, however, a few minor questions and comments which arise:
- There are some of spelling errors which should be addressed, for example,
in line 288 “fall, is the angle between” what is the angle between?
in line 30 “, a subfemtosecond barrier (280 a.s.) with high peak power, ” What does it mean?
- There are some of typographical errors of expressions which should be addressed, for example,
in line 83 Eq. (2),
in line 83 Eq. (40) which differs from their published studies,
in line 120 which differs from their published studies,
- The symbols in the formula and the text should keep the same style.
- The style of representing scalar products should be uniform throughout the text.
However, beyond these minor comments, I feel this paper presents enough interesting and novel content which falls within the scope of Nanomaterials and the interests of its readers to recommend acceptance of this manuscript for publication.
Reviewer 3 Report
In this work, Eseev et al. introduce a theoretical study on ultrafast X-ray (USP) scattering by polyatomic systems.
The novelty of the work consists in the evaluation of the magnetic component of the EM field on the scattering process, which is generally not taken into account by other models.
Authors start from a theory developed by themselves (and correctly cited as a reference in the paper) and particularize it to some cases of interest for which the influence of the magnetic part of the USP cannot be neglected, leading to SHG. For each case, they define a factor depending on the geometric shape of the structure undergoing X-ray sctattering. Among the different results, they interestingly show an X-ray antireflective behaviour of CNTs.
For all of these reasons, the work deserves publication in Nanomaterials, in my opinion.
I only have one concern: I would make the presentation of the mathematical model more accurate. In most cases, some symbols (or variables, or parameters) shown in the different equations or in the text are not adequately defined. In some other cases, they are first introduced in the equations and then defined much later in the text. Finally, some symbols (e.g. α) are used twice for indicating different things.
I recommend the authors to carefully check the model before re-submission, so to make its presentation more accurate.
Minor comments:
Lines 29-31-186 (and others): Replace "a.s." with "as".
Line 31: What do the authors mean with "barrier"? Please clarify.
Line 68: Replace "Plank" with "Planck". Also, specify that it's the "reduced" Planck's constant.
Line 96: Replace "monatomic" with "monoatomic".
Line 108: G and F functions, being crucial for the model, as well as for understanding the results, should be better introduced and described.
Line 166: Radius ring R has already been defined at Line 161.
Round 2
Reviewer 2 Report
The paper was significantly improved and thus may be published in the present form.Reviewer 3 Report
I'm happy with the revised of the paper.
My suggestions and comments have been welcomed by the authors.
The mathematical model and the related equations are now very accurately described.
In my opinion, the paper can be accepted for publication in the present form.